# Social Distance Approximation on Public Transport Using Stereo Depth Camera and Passenger Pose Estimation

**DOI:** 10.3390/s23249665

**Published:** 2023-12-07

**Authors:** Daniel Steven Bell, Philip James, Martín López-García

**Affiliations:** 1School of Engineering, Newcastle University, Newcastle upon Tyne NE1 7RU, UK; philip.james@newcastle.ac.uk; 2School of Mathematics, University of Leeds, Leeds LS2 9JT, UK; m.lopezgarcia@leeds.ac.uk

**Keywords:** social distancing, computer vision, stereo camera, pose estimation, RaspberryPi, public transport, Quantitative Microbial Risk Assessment (QMRA), Transport Risk Assessment for COVID Knowledge (TRACK)

## Abstract

In order to effectively balance enforced guidance/regulation during a pandemic and limit infection transmission, with the necessity for public transportation services to remain safe and operational, it is imperative to understand and monitor environmental conditions and typical behavioural patterns within such spaces. Social distancing ability on public transport as well as the use of advanced computer vision techniques to accurately measure this are explored in this paper. A low-cost depth-sensing system is deployed on a public bus as a means to approximate social distancing measures and study passenger habits in relation to social distancing. The results indicate that social distancing on this form of public transport is unlikely for an individual beyond a 28% occupancy threshold, with an 89% chance of being within 1–2 m from at least one other passenger and a 57% chance of being within less than one metre from another passenger at any one point in time. Passenger preference for seating is also analysed, which clearly demonstrates that for typical passengers, ease of access and comfort, as well as seats having a view, are preferred over maximising social-distancing measures. With a highly detailed and comprehensive set of acquired data and accurate measurement capability, the employed equipment and processing methodology also prove to be a robust approach for the application.

## 1. Introduction

### 1.1. Background

With the worldwide onset of SARS-CoV-2 virus in early 2020, one of the key questions posed by policy makers was what the relative risk of different activities was. This became particularly pertinent as society emerged from lockdowns, and economies were struggling. Early on, the UK government funded a programme of research to understand the (relative) risk of exposure to SARS-CoV-2 on public transport and assess the efficacy of policy measures such as mask wearing, social distancing and limiting occupancy. The transmission of any viral load depends on a multitude of factors, such as how the virus is transmitted (airborne, fomite, etc.). Exposure to pathogens in indoor environments, and in particular in public transport settings, can be modelled using Quantitative Microbial Risk Assessment techniques [1]. These models rely heavily on assumptions around environmental conditions (e.g., temperature, humidity, ventilation) and human behaviour patterns, which were initially poorly understood or backed up by empirical evidence. These included how and where people sat, whether social distancing was possible and how far apart people were from others during the journey, what surfaces they touched and how often they touched their faces and mucous membranes, or how likely passengers would sanitise their hands or wear facemasks. Initial studies using onboard CCTV from transport operators, collected for operational and security purposes, clearly demonstrated that due to camera and image quality and camera placement, they were less than ideal for all but the coarsest measures. This study addresses some of the quality issues with onboard cameras through the deployment of specialist, low-cost cameras in public transport.

One way to monitor passenger activity is with the use of CCTV. CCTV systems on public transport have long remained an effective approach for increasing safety and security for passengers and operators alike. It is for this reason that many vehicles come pre-installed with CCTV systems as standard, directly from the manufacturer. Despite this, in order to minimise data storage expenditure and device costs, camera quality typically remains low in terms of pixel and temporal resolutions. Furthermore, in order to capture as much of the surrounding environment as possible, pre-installed camera devices can employ heavily distorting lenses. The number of camera devices and positioning can also be very unpredictable—not always reducing ‘blind spots’ when other vehicle design constraints and maintenance ability take precedence. Lastly, as existing CCTV systems employ standalone monocular camera devices, any spatial information within the object space is difficult to be accurately reconstructed. Therefore, when advanced analysis of footage is concerned, such as understanding detailed passenger behaviour in respect to social distancing, surface contact, and hand-to-face actions, existing CCTV systems are generally unsatisfactory for purpose.

The presented research suggests a novel approach for capturing data on public transport for the purpose of social distancing understanding. With use of a single stereo depth camera (Intel RealSense D435i, Intel^®^, Bangkok, Thailand) for spatial data capture, and the subsequent application of computer vision techniques, the precise passenger positioning information which is required for social distancing analytics can be extracted. In doing so, the work aims to bridge the gap between low-cost computer-vision applications in transportation, and computer vision applications for the purpose of social distancing understanding.

### 1.2. Aim

The aim of this study is to investigate the state of social distancing on public transport during the latter stages of UK public lockdown restrictions. While doing so, the efficacy of stereo depth cameras for the purpose of social distancing analytics on public transport will also be assessed. Outcomes will provide insight into technical measurement capability, standard passenger conduct, and social distancing capacity with respect to both individual and group contexts—aiding in the development of infection transmission modelling as well as providing a potential solution for future social distance measurement.

## 2. Related Work

### 2.1. Computer Vision in Transportation

A common use of computer vision (CV) analysis on public transport is for passenger counting [2]. This is often motivated by aiming to understand and ameliorate congestion, enforce safety, and improve route and capacity planning within public transport networks [3]. Where ticketing information is unavailable on free-to-ride bus services, one study [4] installed a RaspberryPi3 computer with low-cost camera device in a vehicle doorway. Here, the MOG2 algorithm was used for foreground estimation, whilst Haar Cascade and HOG algorithms were used for object detection. The use of such classic computer-vision approaches has been found to be much more efficient for passenger detection when employed within public transport environments on low-cost devices [5]. In another method [6], used Artificial Neural Networks (ANNs) across a bus doorway to determine passenger detections and trajectories with the aim of improving safety on alighting and disembarking. The utilised algorithm here is capable of estimating passenger density by number of detections within the confined space and limited camera view, despite the focus of the paper not relating to social distancing. Again, as ANN/CNN processing is more computationally expensive than traditional CV approaches, a more expensive and power-hungry compute device is required to perform person detection in situ. To understand if a passenger is boarding or alighting from a single camera, passenger motion trajectories can be tracked following detection, using such methods as Kanade-Lucas-Tomasi (KLT) algorithm [7] and fuzzy logic controllers [8].

### 2.2. Computer Vision in Social Distancing

Various computer vision techniques can be employed to aid in infection transmission modelling in public transport settings, including person detection for social distance monitoring, facial recognition for mask wearing compliance, and pose estimation for symptom detection [9]. To measure social distancing in the real world, it is necessary to restore depth and scale dimensions, which are lost in image capture. Within a monocular camera setup, this can be achieved by inverse perspective mapping (IPM), where a translation matrix is found between the image plain and real-world by corresponding coordinates [10]. By labelling each person’s position on the ground plane as the IPM transformed midpoint of the bottom edge of detections, real-world Euclidean distances between people can then be measured [11]. Alternatively, this has been achieved with the addition of pose estimation and pedestrian tracking (with low-resolution footage), for improved measurement ability [12]. With knowledge of extrinsic camera parameters such as height, and internal parameters such as focal length, a person’s distance to the camera device can be determined and converted to real-world distances before person-to-person distances are determined [13]. Pose estimation can also be utilised to enhance scene restoration with this approach, allowing for auto-calibration techniques before social distance measurement [14]. Several methodologies are successful in detecting people but fail to perform any mapping between the image plane and object world—instead relying on apparent arbitrary threshold values [15], simple Euclidean distances between two image pixel points [16], or scale based on highly unreliable detected bounding box sizes in relation to expected human dimensions [17,18].

Despite existing published literature on the monitoring of people using computer vision, both on public transportation and for social distancing, studies in which these purposes intersect appear largely unexplored. A system has been proposed in which IoT sensors (GNSS-obtained vehicle location, thermal infrared cameras, and microphones) onboard public transport are utilised to gather and process information on general passenger well-being before aiding passengers in making informed decisions on their journey based on real-time data [19].

## 3. Data Acquisition

The bus used in the study was a Wright StreetLite Door Forward (DF) model (a single decker vehicle with 41 passenger seats). At the time of the study, there were two typical service routes, having approximate journey times of 1 h 30 min and 40 min one-way, respectively.

An Intel RealSense D435i depth camera was positioned in a location that could best maximise information capture, both in terms of micro and macro detail (Figure 1). This relatively small (90 × 25 × 25 mm) and low-powered device is capable of capturing depth information up to 20 m, with a calibrated accuracy of <2% at 2 m. By calculating depths on-board and in real-time from both stereo vision imagers and class 1 Infrared (IR) projector techniques, the RealSense device is robust in terms of capture frequency, precision, and redundancy, should one capture method fail. This also allows for ease of deployment and reduced processing load on the external compute device.

To allow for stereo camera and infrared depth alignment, resolution was limited to VGA format (4:3 aspect ratio at 640 × 480 pixels). Therefore, the optimal position for the depth camera was estimated at approximately halfway along the length and width of the vehicle, looking towards the back where the most seats are visible in a single frame at short range. The camera was placed within a protective custom shield before being securely attached upside-down to a horizontal ceiling panel. Wiring was then strung above ceiling coves directly from the D435i device to the RaspberryPi4 (RPi) compute device (placed within a cabinet above the driver’s cab) using an active USB cable.

The RPi performed several functions: supplying power to the D435i device, initialising the D435i device at start-up, recording progress logs, data processing, and data storage to an external thumb drive in compressed and encrypted formats. Additionally, the RPi controlled the D435i device by specifying capture rate (once every five seconds), as well as instructions to align image and depth data according to the intrinsic camera model. All other imaging tasks were handled by the OpenCV library—compiled on the RPi from source code. As the study did not demand real-time processing of social distancing metrics, computationally heavy tasks, such as passenger pose estimation, were intentionally omitted from the edge computation.

The RPi itself was powered via a 12 v automobile auxiliary power outlet to 5 v USB adaptor and would activate whenever sufficient power was supplied to it—i.e., whenever the vehicle engine and/or battery were engaged and consistently running. A Real Time Clock (RTC) module was preinstalled on the RPi to enable timestamping in data collection, as on-board bus internet connectivity was unavailable.

## 4. Methodology

### 4.1. Pre-Processing and Sorting

As the Intel RealSense D435i device installation was inverted, images and depth matrices were firstly rotated by 180 degrees. Next, MD5 image hashes were generated and compared with one another in order to remove any exact image error duplicates. Both left and right imagers of RealSense D435i have horizontal, vertical, and diagonal depth field of view (FOV) of 74, 62, and 88 degrees, respectively in VGA format, resulting in an invalid depth band around the depth map where there is a non-overlap of data. To eliminate this area, as well as reduce the number of partially visible passengers in view, all images and corresponding depth arrays were cropped to exclude any possible detections within a 30-pixel radius from the image border. In addition to these steps, masks were drawn over the image in specific areas to also exclude any possible detections found to intersect these regions. For example, handrails were masked to prevent any inferred detections behind a handrail, giving a false depth reading. Areas susceptible to fooling the detection model into giving false-positive results were also masked (Figure 2). Furthermore, areas between some handrails (where distances were difficult to measure due to depth shadows, obstructions, and camera depth range), were also masked.

### 4.2. Passenger-to-Passenger Distances

Detections were carried out utilising R-CNN R50-FPN 3× algorithm within Detectron2 framework [20]. A notable benefit of this algorithm is the inclusion of pose estimation—image Keypoint coordinates of 17 body part locations to allow imposed stick-man reconstruction (Figure 3). Keypoints were found for each passenger within each image by running all colour images through the Detectron2 predictor.

Following pose estimation, each image was then analysed to extract nose-to-nose distance measurements between every possible combination of passengers (providing the image included more than one passenger detection). As this study focuses on social distancing as a risk factor for airborne pathogen transmission, nose Keypoints were selected as the most appropriate location for which to measure distances between individuals. Measuring a distance m between individuals consists of several stages:Nose Keypoint image coordinates *u*_0_, *v*_0_ and *u*_1_, *v*_1_ are taken of passengers *P*_0_, *P*_1_ on the colour image to locate depth values *d*_0_, *d*_1_ on the corresponding depth array;To scale depths appropriately in finding passenger nose distance *D*_0_, *D*_1_ to the left camera sensor, depth values *d*_0_, *d*_1_ are multiplied by scale value *S* (intrinsic value determined during capture);Distances *D*_0_, *D*_1_ along with positions *u*_0_, *v*_0_ and *u*_1_, *v*_1_ undergo deprojection to compute the corresponding points *P*_0_(*X*,*Y*,*Z*) and *P*_1_(*X*,*Y*,*Z*) in 3D space relative to the camera. This computation requires known camera intrinsic model: distortion model, distortion coefficients, focal lengths of the image plane, height, and width of the image in pixels, and principle point coordinates;Pythagorean formula was used to determine distance measurement m between *P*_0_(*X*,*Y*,*Z*) and *P*_1_(*X*,*Y*,*Z*).

For the purpose of analysis, measured nose-to-nose distances were then separated into three risk categories depending on their value: less than 1 m (increased risk), between 1 and 2 m (moderate risk), and greater than 2 m (lower risk). The 1 m and 2 m thresholds to describe increased transmission risk have been suggested in numerous modelling approaches [1,21] as well as experimental studies [22] and are broadly in line with the definitions of a contact (e.g., for the purposes of contact tracing) that have been used during the COVID-19 pandemic by the WHO [23], the ECDC [24] and the CDC [25]. Furthermore, a systematic review and meta-analysis concluded that these thresholds are also appropriate for specific application to the transmission risks associated with COVID-19 [26].

Analyses were conducted across all available data to examine several metrics against bus occupation levels: number of detections, total number of risk category instances, typical number of risk category instances per image, as well as individual likelihood of falling into any of the risk categories. Excluding standing areas, the camera covered an area suitable for occupying up to 25 passengers, which for the purpose of passenger-to-passenger distance analysis, would be considered maximum capacity. Finding the approximate risk for individuals was conducted by calculating the mean proportion of times each type of risk occurred per level of occupation, over all images (containing at least 2 people). No distinction was recognised regarding the number of times an individual belonged to a particular risk category, with one such occurrence being enough to qualify an individual to a category. Furthermore, an individual could belong to multiple risk categories simultaneously, providing there were more than two people occupying the bus at varying distances from one another.

### 4.3. Seating Positions

Quantitative Microbial Risk Assessment (QMRA) techniques which have been recently used to estimate exposure to SARS-CoV-2 in public transport settings [1] consider different scenarios with varying levels of bus occupancy and make specific assumptions on seating position choices by passengers throughout a journey. In order to improve the realism of these models, the prior methodology of measuring nose-to-nose distances was expanded to analyse occupied seating positions over time:Map distance plane to bus plan (Figure 4):
A total of 15 corresponding reference points were located on the image plane and a plan diagram of the bus model. Features such as the back of seat rests and handrail points were among reference points used;A distance plane was generated by reducing reference point distances and angles to a horizontal plane, before converting these polar positions to the Cartesian system. Corresponding depths were identified for reference points on the image plane, and polar distances *D*_0_, *D*_1_ determined to the camera (as in Section 4.2). Polar angles *θ* were found by: *θ* = *u*·*HFOV/w*
(1)Perspective transformation relationship *H* was determined between corresponding points on distance and bus plan planes. Least median of squares robust estimation algorithm was employed [27].Passenger polar distances *D*_0_, *D*_1_ and angles *θ* are converted to Cartesian system and are plotted onto bus plan with a perspective transformation utilising *H*.Determine which seat (or aisle) positions are occupied based on if a point is contained within its boundary on the bus plan. Visible rows are labelled A-F from front to back, and columns labelled 1–4 from offside to nearside, with M as notation for middle seat in row F (Figure 4).


## 5. Results

### 5.1. Acquired Dataset

Of the 56 days during which the devices remained on the bus, data were collected on 32 of the days when the bus was operational, and storage on the USB thumb drive remained below maximum capacity. Recording at up to five frames per second (when power delivery was stable), the total number of colour images (with corresponding depths) after post-processing totalled 173,044 (Table 1).

### 5.2. Passenger-to-Passenger Distances

Out of the total number of images collected, 97,948 images contained at least two people that could be used for social distance measurements. Counting the number of passenger detections in each image reveals that (from what is visible to the camera) the bus typically has few people occupying it. As depicted in Figure 5a, there are 25,770 images of the bus at 8% occupation, 21,254 images at 12% occupation, and 16,107 images at 16% occupation. As the level of occupation increases, the number of available images continues to decline—until only seven images are available at 56% occupation. Following this point, no further samples were successfully captured, leaving higher occupations unexplored. Consequentially, an occupation of at and below 12% accounts for almost half of the available dataset.

Figure 5b shows the total number of distance measurements taken per occupation level, for each risk category. For example, at 24% bus occupancy there are 17,507 less than 1m readings collected over all days. Despite the number of possible measurements increasing for each additional passenger according to a triangular number sequence, the lack of available images for higher levels of occupation counteracts this to produce a steady decrease in measurements taken for all categories beyond 24% capacity. Figure 5c shows the typical number of measurements taken in each risk category per image, all of which increase in number along with the total number of possible measurements per level of occupation. For example, at 36% occupation, there are a total of 36 measurements taken between nine passengers, 12.8 of which are typically between 1 and 2 m, 4.6 are at less than 1 m, and the remaining 18.6 are in the greater than 2 m category.

Figure 5d depicts the likelihood of an individual falling into any of the risk categories, from at least one other person on the bus, per level of bus occupation, at any one point in time. For example, at 36% occupancy, there is roughly a 67% chance an individual will be within 1 m of at least one other person. Furthermore, it is very likely (>92%), an individual at this occupancy will also be within 1–2 m from at least one other person. As expected, categories tend toward 100% likelihood as occupation increases—in particularly those greater than 2 m and 1–2 m categories, as there is little chance of avoiding these distances well before the bus reaches maximum capacity. The less than 1 m category follows a similar trajectory but does not reach beyond a 78% likelihood, even at occupancies above 50%. Around this mark, the less than 1 m category also appears to level off, and also decreases slightly (by 1.5%) at the highest recorded level of occupancy. This was unanticipated, with any levelling off expected only toward nearing a 100% likelihood—as clearly demonstrated in the greater than 2 m category—and most probably stems from limitations of the used method, in both accurately detecting and measuring distances between passengers in a crowded environment. The less than 1 m category does not increase as dramatically as the other two categories, which is evidence that passengers generally try to space themselves from one another. In addition, the greater than 2 m category does not immediately reach 100% likelihood at lower occupancies, indicating individuals possibly choose to sit within closer proximity to one another and in groups, even when there are few people on the bus. When there are only two individuals within view, there is a 15% chance they will choose to sit closely/next to one another, and a 29% chance they will be within 1–2 m from one another. Risk category calculations are independent, meaning the chosen individual can fall into multiple categories should there be more than one other passenger present. Consequently, these results should be interpreted as a very best-case scenario, with the reality being several simultaneous exposures of varying distance for a passenger.

### 5.3. Seating Positions

Figure 6 shows the number of passenger counts over all days. In total, 395,693 detections were allocated to seats from a collection of 126,584 images containing one or more passengers. Furthermore, Figure 7 represents these values in terms of percent of time occupied. Percent of time occupied ranges widely, from only 3.52% in labelled position C3, to 50.10% in adjoining seat C4. It is evident that window-seats on both sides are generally the most popular, with four instances (A4, B1, C1, and C4) counting in excess of 40,000 instances each—being occupied greater than 32% of the time. Conversely, aisle-seats are less popular, being occupied at most, only 17% of the time at C2. As passengers only use the aisle area when travelling to and from their chosen seat, time spent here is relatively low at 5.19%. The only visible seat on the back row (FM) falls somewhere in the middle of the pack, being occupied 12.98% of the time.

Using the seating position information, event plots were drawn per day representing which seating positions were occupied over time. Figure 8 demonstrates seating occupation over four consecutive days. In these, bus downtimes can be identified by large gaps in the graph for every seat, as well as rush hour periods by more seats being occupied during particular periods.

## 6. Discussion

### 6.1. Depth Quality

The point cloud model depicted in Figure 9 can be used to understand the quality of depth modelling and accuracy. In this visualisation, seats, handrails, and the aisle can clearly be identified, as well as some surrounding vehicle structure such as coving and aisle steps. Due to the opacity of windows failing to reflect enough IR information back to the D435i device, much of the detail in these sections are lost, distorted, or heavily elongated as exterior features are detected. However, such points do not pose an issue for conducting analyses within the bus interior and have therefore mostly been removed from the visualisation. As expected, the D435i device performs best in X and Y dimensions (Figure 9b), but somewhat reduces in quality in the Z (depth) dimension. Despite appearing typical in Figure 9b, some artifacts can be examined in side-looking plots Figure 9a,c, where handrails appear to merge with seats. Passenger measurements taken on the edge of masked handrails may have been subject to the effects of this. In side profiles, the closest, nearside handrail can also be seen to take on a waved form in the Z dimension—potentially from the effects of IR interacting with a specular surface material. Moving towards the back of the vehicle, points in the Z dimension continue to degrade with increased objects, noise and blind spots.

Collecting depth information in such a confined space will always pose a challenge. However, to improve overall visibility and reinforce all point positioning, several synchronised and calibrated devices can be employed in different positions and orientations—made feasible by the low-cost, low-power, simple installation, and scalable design of the employed system.

### 6.2. Experimental Outcomes

The rapid decrease in and unavailability of detections in higher levels of occupation is likely a combination of several factors—detection accuracy/limitations, passenger footfall, and passenger awareness. In an increasingly busy and obstructed scene, detections are made more difficult for the algorithm. Furthermore, the relatively low sensor resolution (standard VGA) further exacerbates this issue, especially in detecting passengers towards the back of the vehicle. It may also be the case that there are fewer instances of the bus being busy simply because it is it not typically a busy service. At the time of study lockdown restrictions were in the process of being eased, with non-essential businesses and public spaces being reopened. This gives reasonable assumption that public confidence in travelling on public transport may have remained low, therefore reducing occupancy. The Department for Transport statistics confirm this, suggesting weekday bus usage (outside of London) was at around 75% of pre-COVID levels during the period [28]. Finally, it may also be the case that, being aware of the camera, some passengers may have chosen to avoid seating/standing positions within view. In any case, the number of distance measurements peaked at 24% capacity.

As the number of passengers increase, at a greater rate so does the number of possible distance measurements which can be taken. Moreover, as the 1–2 m category has a valid area three times that of the less than 1 m category, increasingly more measurements will likely be taken in the 1–2 m range as capacity increases. Results reflect this, with the total number of 1–2 m instances accounting for 35% of all measurements taken, in comparison to that of 13% for the less than 1 m category. The remaining 52% of measurements taken can be attributed to the greater than 2 m category. However, these physical constraints only have limited effect in reducing individual likelihood of sharing in one of these groups. Despite best efforts of individual passengers to space themselves from one another (with some evidence this is possible), results indicate that with an increasing level of occupation it is very difficult to not be within 2 m of another passenger within the captured dataset. Given the confined space within the bus, at higher levels of occupancy it is obvious that this trend should continue in spite of the apparent trajectories of the 1–2 m and, more so, the less than 1 m categories derived from the captured dataset. Given these chances over the timeline of a journey, rather than at a single point in time, makes social distancing for a sustained period incredibly difficult, even at relatively low capacities.

By analysing seating positions, some evidence is revealed on how passengers naturally space themselves from one another. With window seats being more popular than aisle seats, passengers are typically at either side of the bus, with the combined width of two seat columns and aisle as available distancing space. However, it is not evident whether any such trend also exists when considering choice seating positions between rows, which could also have an impact on infection transmission risk depending on ventilation, windows opening and the specific airflow dynamics within the bus during the journey. Choice seating between rows could also have an impact on fomite transmission, with passengers seating in the final rows possibly contacting more surfaces on their way in and out of the bus. As well as being by the window, the most popular seat is found to have several other special characteristics which may explain why it is chosen so frequently. It is on the nearside of the vehicle, away from oncoming traffic and on the same side as the passenger doorway. The most popular seat is also on the first row which is raised above all previous rows—giving the passenger an unobstructed view of the front of the vehicle. Lastly, a handrail is conveniently placed for right-handed passengers to pull themselves into this seat and the seat adjacent when finding a position. All analyses here point towards convenience, comfort, and experience as being the most valued concerns of bus passengers, at least in the lower levels of occupations where most of the detection data resides. For reasons also pertaining to personal comfort and safety, prior research suggests the primary concern of many passengers is to avoid sitting directly beside another individual when travelling [29]. Subsequently, any achieved social distancing (or lack of), is most likely a by-product of these collective choices. It may be that having accepted the inevitable social distancing risks that come with being in a confined space, passengers instead prioritise other factors which will better improve their journey, and which can be better controlled. Conversely, at greater capacities, and with a decreased choice in seating positions, the importance of social distancing could again increase above that of accessibility, comfort, and enjoyability.

### 6.3. Employed Approach

The most challenging aspect of the employed method was in data acquisition. Most notably, installing the required packages and software to the RPi device in order to correctly interface with the camera and to consistently extract, process, and store resulting data. As the devices were installed remotely with no form of wireless/web communication for transfer of data, metadata, or logs, a high level of robustness was critical. The system was stress tested beforehand to ensure long periods of accurate data collection, as well as restart reliability following a loss of power or software crash. Nevertheless, on first inspection of the data post installation, it was found an issue had arisen with saving data to the external drive. The formatting type of the external drive, in conjunction with the long naming convention used in saving files, meant disk space was utilised inefficiently and had filled much quicker than expected. Having the ability to connect to the RPi over a wireless network would ensure much more accessible monitoring, maintenance, quality assurance, and accurate time-keeping ability, as well as potential data transfer for backup redundancy. Heavy computational tasks (e.g., detection) and following processes, were conducted after the data acquisition stage, and therefore proved relatively simple to develop and execute. In the interest of combining all processing stages to enable for real-time monitoring and analytics on the edge, this could be achieved by improving processing efficiency whilst employing more powerful hardware, along with prior mentioned remote accessibility.

Using nose Keypoint positions to determine passenger location in the depth scene, proved as a suitable technique for measuring the distances between passengers and determining occupied seating positions. Other computer vision techniques for measuring social distancing rely upon bounding boxes for person detection [10,11,13], and inverse projection mapping/level floor surfaces for spatial reconstruction [10,11,12,14]. As opposed to these, the employed method in the presented research excels in that multiple positions of a person can be identified without any prior spatial knowledge or required surface conditions in the object space—allowing for immediate and detailed social distancing analysis to be performed. Scene complexity, the quality of capture, and the detection algorithm used have the most significant impact on the ability to accurately identify and precisely locate Keypoint positions within the image space. By further training the Detectron2 R-CNN R50-FPN 3x algorithm for the scene of capture and camera image quality, an increase in true positive observations and a decrease in false positive and false negative observations may be observed.

The Wrightbus StreetLite DF model was selected due to its availability by the service provider. In 2020, the number of new Wrightbus bus registrations in the UK only accounted for 1.3% of the total number of new bus registrations [30]—of which, not all may have been the model used in this study. If the statistics of 2020 represent the wider use of buses in the UK, then it can be assumed relatively few services operate with use of a StreetLite DF model. Despite this, the sample data and results collected can still be considered representative of the typical conditions found on many other bus models. The four-seat rows, separated by an aisleway, within the studied proportion of the vehicle is characteristic of layouts found in many single-decker models in use around the UK. This includes an increase in elevation towards the rear of the vehicle for engine placement. Models which do not follow this setup may influence results somewhat, based on number of seating positions, vehicle size, and service popularity. However, with limitations on vehicle size and the need to maximise passenger capacity, any variation in results is not expected to vary greatly. In terms of application, the techniques explored in this study can be utilised in any other bus model with minimal setup, providing the camera can be positioned with an adequate, unobstructed view of seating areas.

## 7. Conclusions

In this paper, related work was first explored for the use of computer vision techniques in both public transportation and social distancing. Machine learning techniques were a prominent method for first detecting people before additional algorithms were applied for accurate counting, tracking, and geometric restoration. However, published work that combines both CV in transportation and CV in social distancing is so far limited in number.

The procedures taken in data acquisition were outlined. This includes choice of hardware, onboard capture, and processing tasks, as well as installation steps. The employed system proved to be low-cost, low-power, highly robust, scalable, and capable of capturing suitable information for passenger pose estimation, passenger-to-passenger distance estimation, and seat occupation, despite the physical constraints of the bus. This approach can be recommended in the future, both within and outside of a public transport setting. In either case, maximising visible view space for the camera device(s) should be emphasised—likely providing even greater acquired spatial accuracy and person detection capability—to reduce a skewed set of data over all levels of bus capacity.

Methodological approaches were explained, outlining the use of the Detectron2 detection framework for pose estimation and subsequent Keypoint transformation to 3D space using associated depth information gathered from the Intel RealSense D435i camera device. From this, passenger-to-passenger distances and seating potions were then determined, along with statistical analysis for understanding group and individual social distancing risks in three categories—less than 1 m, between 1 and 2 m, and greater than 2 m.

As expected with being in a confined space, the results proved social distancing is difficult to achieve even at low vehicle occupancy—with a 93% chance of being within 1–2 m from at least one other passenger and a 77% chance of being within less than 1 m from another passenger at 50% occupancy. A limitation of this study is that the duration of contact was not considered. It is very likely that infection transmission risks would further increase when factoring in contact duration. An analysis of passenger seating positions revealed that factors such as comfort, ability to access, and enjoyment of the ride potentially had a greater influence on choice than social distancing, especially at lower levels of occupancy. Further research and data capture are required to understand if this is also the case at greater levels of vehicle occupancy or if passenger values change with increased risk.

The outcomes gained in this work reveal the state of social distancing within a slice of public transport journeys during semi-lockdown conditions in the UK—providing valuable results and analyses for the enhancement of infection transmission modelling. Furthermore, a method for the acquisition and processing of social distancing data has been explored, building upon available solutions for future pandemic responses within the transportation sector.

## Figures and Tables

**Figure 1 sensors-23-09665-f001:**
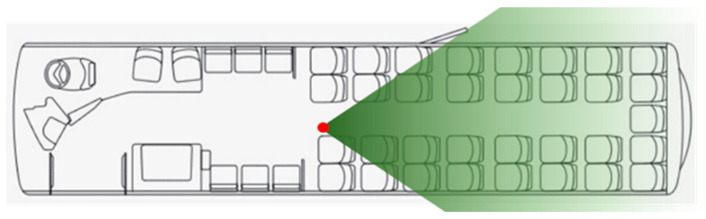
Wright StreetLite DF plan with camera position (red) and viewing orientation (green).

**Figure 2 sensors-23-09665-f002:**
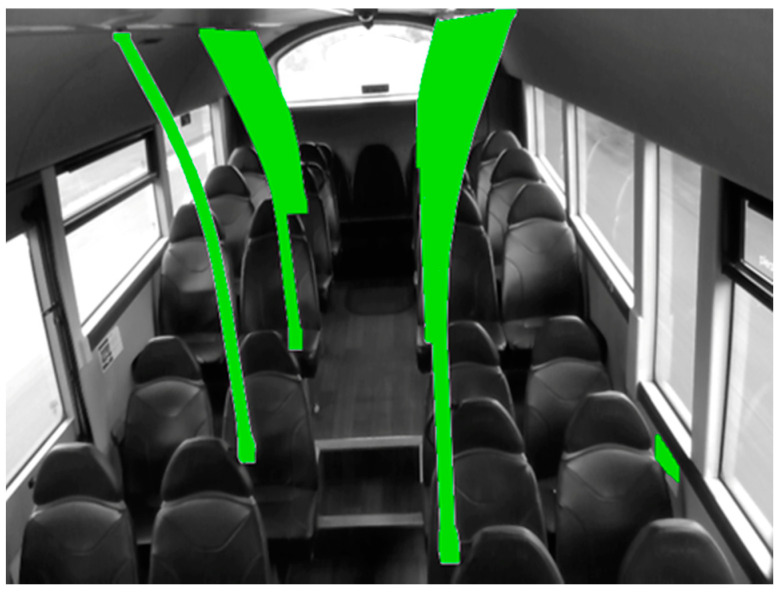
Camera view: Masked areas (green) to omit regions which would give false depth readings or could potentially fool the detection algorithm.

**Figure 3 sensors-23-09665-f003:**
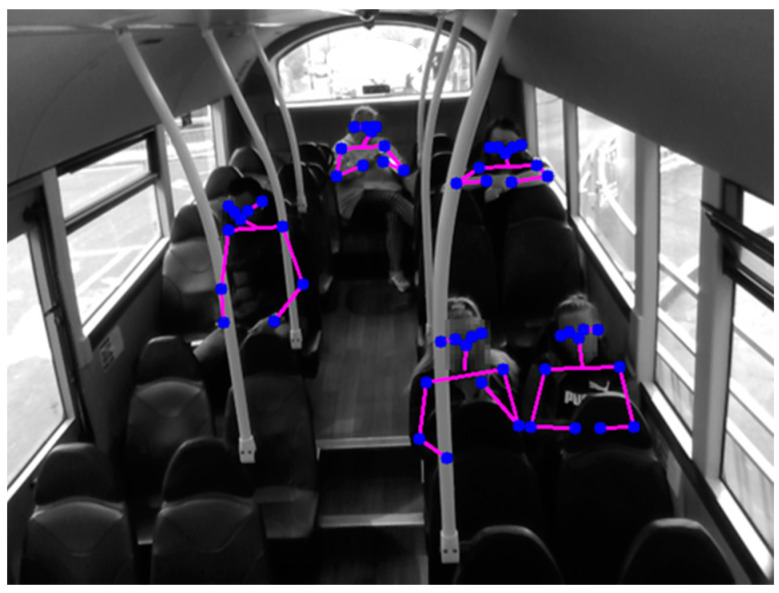
Camera view: Pose estimation detections of five anonymised passengers on the bus, showing eye, ear, nose, shoulder, elbow, and wrist Keypoints as blue dots, and pink lines connecting Keypoints.

**Figure 4 sensors-23-09665-f004:**
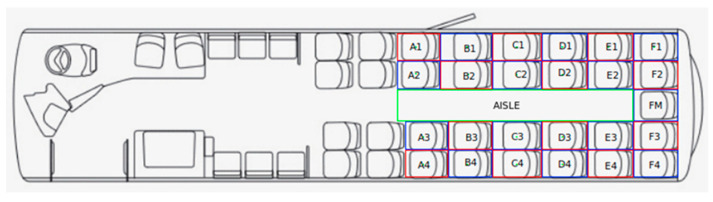
Labelled Seating Positions and Boundaries on Bus Plan. Several seats (A1, D2, D3, E2, E3, F1, F2, F3, F4) were omitted from calculations as passenger detections in these locations were inaccurate due to lack of visibility from unfavourable camera angle, obstructions, and degrading depth and image quality towards the back of the vehicle.

**Figure 5 sensors-23-09665-f005:**
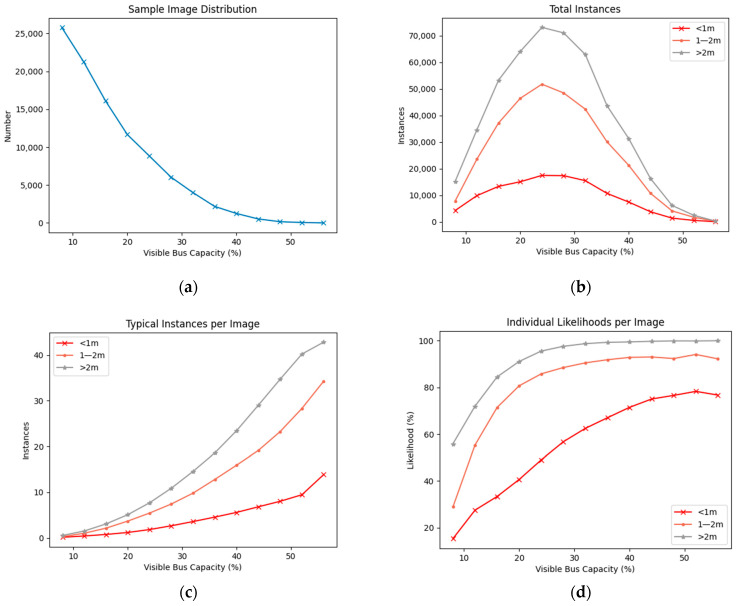
Passenger to passenger distance analysis: (**a**) Sample Image Distribution. As occupation increases, the number of available sample images (containing the corresponding number of detections) decrease. No samples were captured above 56% occupancy; (**b**) Risk Instances per Distance Category. The number of achieved distance measurements increase with greater occupancy but taper off towards higher occupancy due to lack of available samples; (**c**) Typical Number of Risk Instances per Image, reveals the distribution of risk categories over an increasing number of total measurements (as occupation increases); (**d**) Likelihoods of Individual Falling into a Risk Category. As occupancy increases, the risk of an individual being 1–2 m or <1 m from at least one other passenger greatly increases.

**Figure 6 sensors-23-09665-f006:**
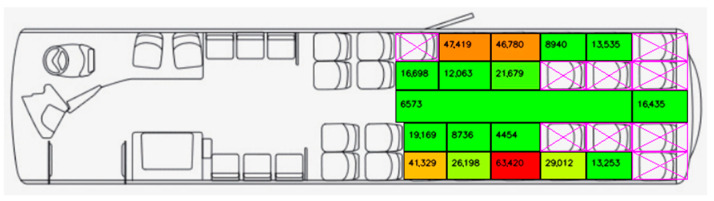
Number of Detections per Seat, where green represents less frequently occupied and red represents more frequently occupied.

**Figure 7 sensors-23-09665-f007:**
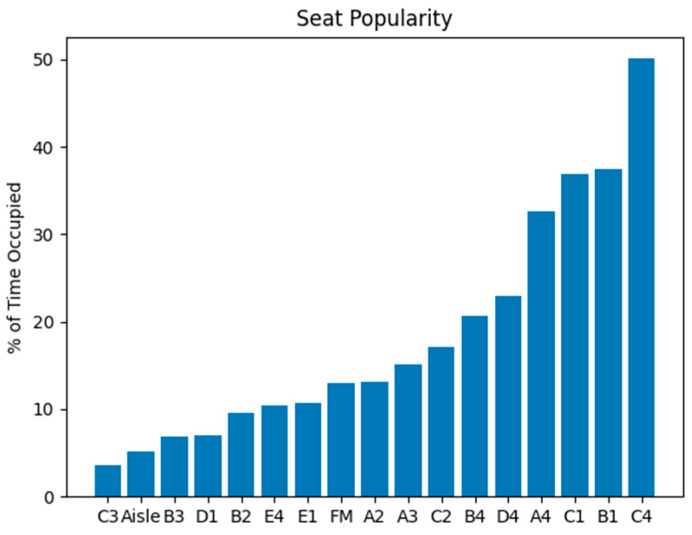
Seat Popularity by Percent of Time Occupied. Seating position C4 is the most popular choice for passengers, being occupied 50% of the time, while adjoining seat C3 is the least chosen.

**Figure 8 sensors-23-09665-f008:**
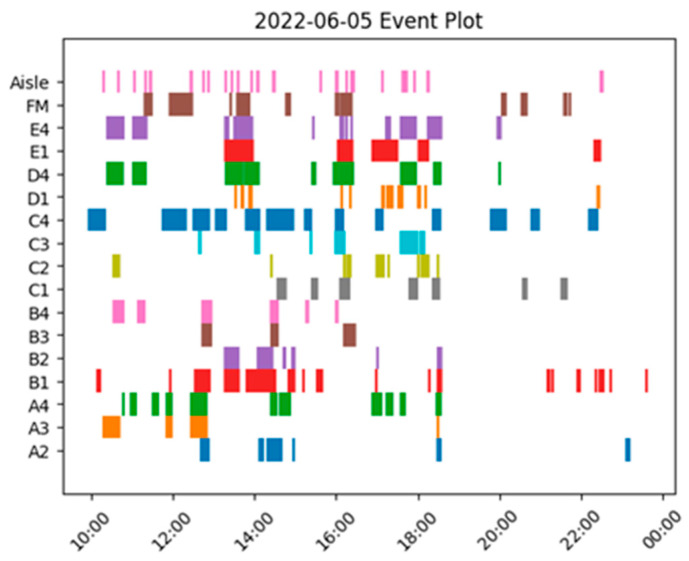
Event Plot for a single day, depicting which seats were occupied over the course of a day. Colours uniquely representative of each seating position. From such graphs and supporting data, choice seating patterns can be analysed for better transmission modelling.

**Figure 9 sensors-23-09665-f009:**
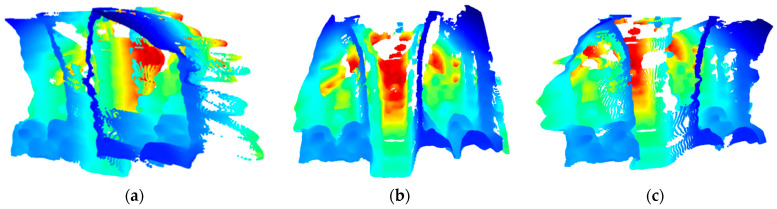
Point cloud visualisation from three perspectives, on a Hue, Saturation, Lightness (HSV) colourmap, with blue representing closer distances and red representing further distances from the camera, respectively. Point positions averaged from 89 frames (with no people in view) using median: (**a**) Perspective from front-nearside towards back-offside; (**b**) Camera perspective—centre position looking down aisle; (**c**) Perspective from front-offside towards back-nearside.

**Table 1 sensors-23-09665-t001:** Acquired data: Collected Data. From the 406,764 detections achieved, a total of 918,075 distance measurements were taken, and 395,693 seat positions were successfully allocated (97% of detections).

Category	Number
Days	32
Images/depth arrays	173,044
RCNN R 50 detections	406,764
Passenger–passenger distance measurements	918,075
Seat/aisle allocations	395,693

## Data Availability

Samples collected and generated in the analysis process contain personal information and as such cannot be published nor made available to interested parties. Aggregated statistics resulting from the work are included in the paper body.

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
