# Peer review of "Social Distance Approximation on Public Transport Using Stereo Depth Camera and Passenger Pose Estimation"

_sensors, 2023, doi:10.3390/s23249665_

Round 1
Reviewer 1 Report
Comments and Suggestions for Authors
The main research content of the article is the use of computer vision techniques to measure social distancing on public transport. The article describes a low-cost depth-sensing system deployed on a public bus to gather data on passenger habits in relation to social distancing. This study provides valuable insights into the challenges of balancing enforced guidance/regulation during a pandemic with the necessity for public transportation services to remain safe and operational.
The structure of this paper is basically reasonable and scientific, with clear logical thinking, accurate expression of viewpoints, fluent language, reasonable experimental methods, and references that meet the requirements of the topic. But the method used lacks significant innovation.
1. Representativeness of sample data: The article mentions the use of a specific model of bus, but does not specify how to select this model and whether it represents other types of buses. The representativeness of the sample is crucial for the general applicability of the study.
2. The article uses the Detectron2 detection framework for pose estimation, which is based on an open source project and does not mention any areas for improvement.
3. During the epidemic, passengers' choice of seats is largely related to personal preferences, psychological characteristics (such as being further away from people who do not wear masks), and interpersonal relationships. The author needs to conduct a thorough investigation and supplement.
Comments on the Quality of English Languageno
Author Response
Many thanks for your comments.
- New text added to discuss the representativeness of the bus model (484-499).
- Suggested improvements for the used pose estimation model and comparison with other methods are discussed (470-483).
- Supporting research in personal preferences, psychological and interpersonal relationships on public transport is limited, and it is not possible to thoroughly investigate with further experimentation. New text and reference (441-443) support the suggestions made regarding passengers choice seating.
Reviewer 2 Report
Comments and Suggestions for Authors
The manuscript is very well written and easy to follow. It describes an exciting solution.
In the introduction, I recommend adding a paragraph with an explicit explanation of the novelty of the proposed solution and what is the main contribution concerning the existing research.
In section 4.2, where the authors have explained how they defined the thresholds (1m, 1-2m, and >2m), some sources are not available, such as [23]. Furthermore, the reference [21] is from 2018. Is it better to include more recent studies with facts about the transmission ability of the COVID-19 virus?
Comments on the Quality of English Language
There are some typos, e.g.,
- Line 166: imagers -> images
- Line 172: boarder -> border
- Line 208: missing "was"
Author Response
Many thanks for your comments.
- Added paragraph for research novelty (65-71).
- Reference [23] is available to me. However have added a source link for [24] and updated link for [25]. Added in additional text and reference related to specific COVID-19 transmission ability in lines (222-224).
Comments on the Quality of English Language
- Use of ‘imagers’ relating to the camera sensors not the images they produce.
Reviewer 3 Report
Comments and Suggestions for Authors
This article proposes a low-cost deep sensing system deployed on buses as a means to approximate social distance measurement and study passenger habits related to social distance. The results indicate that for individuals, social distance on this form of public transportation is unlikely to exceed the occupancy threshold of 28%, with an 89% chance of being within 1-2 meters of at least one other passenger, and a 57% chance of being within 1 meter of another passenger at any point. We also analyzed passengers' preferences for seats, clearly indicating that for typical passengers, seats that are easy to enter, comfortable, and have a view of the scenery are more popular than measures to maximize social distance. With a highly detailed and comprehensive set of data collection and precise measurement capabilities, the equipment and processing methods used have also proven to be a robust application method. But there are also some issues:
1. The "introduction" section of this article explains that the content of the background section is redundant and unclear, and the structure is slightly abrupt. The introduction suggests that the author condense it into three key points, highlighting innovation. (For details, please refer to and refer to " Minimal Color Loss and Locally Adaptive Contrast Enhancement")
2. The images of Fig2 and Fig3 are not clear. Please provide high-resolution images.
3. The data collection section is too long, and it is recommended that the author refine its key points and structure. (For details, please refer to and refer to " Weighted Wavelet Visual Perception Fusion").
4. This paper is mainly to approximate the social distance measurement and study the passenger habits related to social distance. The content of COVID-19 epidemic need not appear many times as an example to introduce suggestions.
5. The method section of the article lacks a comparison with the latest method and should be added. In the discussion section of this article, a more detailed explanation should be provided. (For details, please refer to and refer to " piecewise color correction and dual prior optimized contrast enhancement").
Comments on the Quality of English LanguageThis article proposes a low-cost deep sensing system deployed on buses as a means to approximate social distance measurement and study passenger habits related to social distance. The results indicate that for individuals, social distance on this form of public transportation is unlikely to exceed the occupancy threshold of 28%, with an 89% chance of being within 1-2 meters of at least one other passenger, and a 57% chance of being within 1 meter of another passenger at any point. We also analyzed passengers' preferences for seats, clearly indicating that for typical passengers, seats that are easy to enter, comfortable, and have a view of the scenery are more popular than measures to maximize social distance. With a highly detailed and comprehensive set of data collection and precise measurement capabilities, the equipment and processing methods used have also proven to be a robust application method. But there are also some issues:
1. The "introduction" section of this article explains that the content of the background section is redundant and unclear, and the structure is slightly abrupt. The introduction suggests that the author condense it into three key points, highlighting innovation. (For details, please refer to and refer to " Minimal Color Loss and Locally Adaptive Contrast Enhancement")
2. The images of Fig2 and Fig3 are not clear. Please provide high-resolution images.
3. The data collection section is too long, and it is recommended that the author refine its key points and structure. (For details, please refer to and refer to " Weighted Wavelet Visual Perception Fusion").
4. This paper is mainly to approximate the social distance measurement and study the passenger habits related to social distance. The content of COVID-19 epidemic need not appear many times as an example to introduce suggestions.
5. The method section of the article lacks a comparison with the latest method and should be added. In the discussion section of this article, a more detailed explanation should be provided. (For details, please refer to and refer to " piecewise color correction and dual prior optimized contrast enhancement").
Author Response
Many thanks for your comments.
- Added paragraph for research novelty (65-71).
- Images in Fig2 and Fig3 are produced by the low-resolution device used in the study, and so high-resolution is not available.
- Although the techniques presented can be applied and modified to fit social distancing in any context, this research has been conducted specifically within the context of COVID-19, with known properties of this virus used to define distance/risk thresholds. The research was also funded out of a response to COVID-19.
- A comparison with other methods has been added to the discussion section (472-478) and discussion section extended (484-499).
Round 2
Reviewer 1 Report
Comments and Suggestions for Authors
The author has made a comprehensive reply and revision to the opinions raised, and I have no other comments.
Comments on the Quality of English LanguageThere are some punctuation errors in the article, such as 383 lines.
Reviewer 3 Report
Comments and Suggestions for Authors
All my concerns have been well addressed. The manuscript can be recommended for publication.
Comments on the Quality of English LanguageAll my concerns have been well addressed. The manuscript can be recommended for publication.